# HBV preS Mutations Promote Hepatocarcinogenesis by Inducing Endoplasmic Reticulum Stress and Upregulating Inflammatory Signaling

**DOI:** 10.3390/cancers14133274

**Published:** 2022-07-04

**Authors:** Wenbin Liu, Shiliang Cai, Rui Pu, Zixiong Li, Donghong Liu, Xinyu Zhou, Jianhua Yin, Xi Chen, Liping Chen, Jianfeng Wu, Xiaojie Tan, Xin Wang, Guangwen Cao

**Affiliations:** 1Department of Epidemiology, Second Military Medical University, 800 Xiangyin Rd., Shanghai 200433, China; wenbinl_lxb@163.com (W.L.); csljair@163.com (S.C.); purui6136@163.com (R.P.); lzx1989@126.com (Z.L.); zhouxy0428@163.com (X.Z.); hawkyjh163@163.com (J.Y.); arhelion@163.com (X.C.); liping-cat@163.com (L.C.); xjtan2020@smmu.edu.cn (X.T.); 2Department of Liver Cancer Surgery, Third Affiliated Hospital, Second Military Medical University, Shanghai 200433, China; liu783011725@foxmail.com; 3Department of Pathology, Xijing Hospital, Xi’an 710032, China; kobewoo1989@163.com; 4Laboratory of Molecular Cell Biology, Institute of Biochemistry and Cell Biology, Chinese Academy of Sciences, Shanghai 200433, China; wangx336@umn.edu.cn

**Keywords:** hepatitis B virus, preS mutation, carcinogenesis, inflammation, STAT3

## Abstract

**Simple Summary:**

Viral mutations at the preS region of hepatitis B virus (HBV) significantly increase the risk of developing hepatocellular carcinoma (HCC). Compared to HBV preS deletion, the oncogenic effect of preS combo mutation has rarely been investigated. With a cohort including 2114 subjects, we demonstrated that preS combo mutations G2950A/G2951A/A2962G/C2964A and C3116T/T31C significantly increased the risk of HCC in patients without antiviral treatment, whereas preS2 deletion significantly increased the risk of HCC in patients with antiviral treatment. The prevalence of C3116T/T31C (43.61%) was higher than preS2 deletion (7.16%). By using *Sleeping Beauty* mouse models and in vitro experiments, we found G2950A/G2951A/A2962G/C2964A, C3116T/T31C, and preS2 deletion promoted hepatocarcinogenesis by increasing levels of inflammatory cytokines, activating STAT3 pathway, enhancing endoplasmic reticulum stress, and altering gene expression profiles in inflammation- and metabolism-related pathways. These results suggest that preS combo mutations G2950A/G2951A/A2962G/C2964A and C3116T/T31C had similar oncogenic effects of preS2 deletion and should also be monitored.

**Abstract:**

This study aimed to elucidate the effects and underlying mechanisms of hepatitis B virus (HBV) preS mutations on hepatocarcinogenesis. The effect of the preS mutations on hepatocellular carcinoma (HCC) occurrence was evaluated using a prospective cohort study with 2114 HBV-infected patients, of whom 612 received antiviral treatments. The oncogenic functions of HBV preS mutations were investigated using cancer cell lines and *Sleeping Beauty* (*SB*) mouse models. RNA-sequencing and microarray were applied to identify key molecules involved in the mutant-induced carcinogenesis. Combo mutations G2950A/G2951A/A2962G/C2964A and C3116T/T31C significantly increased HCC risk in patients without antiviral treatment, whereas the preS2 deletion significantly increased HCC risk in patients with antiviral treatment. In *SB* mice, the preS1/preS2/S mutants induced a higher rate of tumor and higher serum levels of inflammatory cytokines than did wild-type counterpart. The preS1/preS2/S mutants induced altered gene expression profiles in the inflammation- and metabolism-related pathways, activated pathways of endoplasmic reticulum (ER) stress, affected the response to hypoxia, and upregulated the protein level of STAT3. Inhibiting the STAT3 pathway attenuated the effects of the preS1/preS2/S mutants on cell proliferation. G2950A/G2951A/A2962G/C2964A, C3116T/T31C, and preS2 deletion promote hepatocarcinogenesis via inducing ER stress, metabolism alteration, and STAT3 pathways, which might be translated into HCC prophylaxis.

## 1. Background

Cancer is the first leading cause of premature death in 57 countries [1]. In China, the first leading cause of cancer death in residents younger than 65 years is primary liver cancer [2]. Hepatocellular carcinoma (HCC) contributed 75–85% of primary liver cancer cases globally but 93.0% in China [3,4]. A total of 84.4% HCC in China are caused by Chronic Hepatitis B virus (HBV) infection [4]. Compared to other HCC-related etiological factors, chronic HBV infection is associated with a 10-year earlier onset and the more aggressive nature of HCC [4]. During HBV-induced hepatocarcinogenesis, HBV evolves gradually, which is partially induced by cytidine deaminases [5,6]. HBV genotypes B and C, the two major genotypes endemic in China, have distinct mutation patterns, especially in the preS region of HBV genome [7]. HBV preS and S gene encode three forms of surface proteins: large (preS1/preS2/S), middle (preS2/S), and small (S) surface antigen, which are directly exposed to immune system. These HBV envelope proteins are synthesized in endoplasmic reticulum (ER). Some HBV mutations, especially those accumulated in the preS gene (preS1, nt.2848–nt.3204; preS2, nt.3205–nt.154), which are selected by host immunity, are significantly associated with increased risks of HCC occurrence and recurrence [8,9,10,11,12]. Among these HCC-related preS mutations, G2950A, G2951A, A2962G, and C2964A are exclusively evident in genotype C HBV; whereas HBV preS deletion is mostly occurring in HBV genotype C [9].

The mechanisms by which the preS mutations improve hepatocarcinogenesis remain poorly understood. Current studies mainly focus on HBV preS deletion, especially the preS2 deletion, rather than single or combo nucleotide substitutions within HBV preS region. The preS deletion-induced mutant envelope proteins accumulate in ER, leading to impaired secretion of HBsAg and ER stress. Experimental data from transgenic mice and HCC cells demonstrate that HBV preS deletion activates carcinogenic signaling pathways in an ER stress-dependent way [13,14,15]. The preS deletion induced ER stress also leads to calcium overload in mitochondria, reduction of ATP production, and liver fibrosis, thereby promoting HCC [16]. However, the effects and underlying mechanisms of combo preS mutations on hepatocarcinogenesis are rarely investigated. Antiviral treatments with nucleos(t)ide analog (NA) and/or interferon α (IFNα) can reduce the risks of HCC occurrence and recurrence, especially in those with HBV mutations in the core promoter region of the HBV genome [17,18]. The prophylactic effect of antiviral treatment on HCC in those with the preS mutations remains unknown. In this study, we evaluated the effects of the preS deletion and combo preS mutations on the occurrence of HCC in a cohort of HBV-infected patients. The oncogenic effects of the preS mutations were then validated by using *Sleeping Beauty* (*SB*) transposon system to deliver the preS1/preS2/S mutants into the livers of fumarylacetoacetate hydrolase (*Fah*)-deficient mice. Molecules regulated by the preS mutations were identified through gene expression profiling analysis. This study helps elucidate the mechanisms by which the preS mutants promote carcinogenesis and reveals potential prophylactic and therapeutic options for HBV-related HCC.

## 2. Methods

### 2.1. Study Population Involved in the Prospective Cohort Study

The cohort study conformed to the ethical guidelines of the 2000 Declaration of Helsinki and was approved by the ethics committees of the Second Military Medical University. The HBV-infected patients admitted to the Department of Infectious Disease, the second Affiliated Hospital of this university during August 1998 to December 2007, were invited to participate in this cohort study. This study launched in January 2008 and initially enrolled 2512 patients who fulfilled the following criteria: (i) they were adults with chronic hepatitis B (seropositive for HBsAg and detectable HBV DNA for more than 6 months) and had not yet received antiviral treatment, (ii) they were willing to provide at least 10 mL peripheral blood before antiviral treatment or at first admission, and (iii) they provided written informed consents. Patients who had decompensated liver function or other forms of liver disease, including alcoholic hepatitis, drug-induced liver injury, nonalcoholic fatty liver disease, autoimmune liver disease, decompensated liver cirrhosis, HCC, or infection of hepatitis A virus, hepatitis C virus, hepatitis D virus, hepatitis E virus, human immunodeficiency virus, and/or Treponema pallidum were excluded from this study. The diagnostic criteria of chronic HBV infection, liver cirrhosis, and HCC, were the same as previously described [18]. The baseline demographic and clinical data were extracted from their medical records. Biochemistry tests for the parameters of liver function, antibodies, and platelet count were performed on the first day of hospital admission. All participants were self-reported Han Chinese.

### 2.2. Antiviral Treatment and Follow-Up

Patients were recommended to receive antiviral treatment if (i) they had cirrhosis with detectable HBV DNA and/or (ii) they had elevated levels of serum alanine aminotransferase (ALT) (twice more than the upper limit of normal, ULN) and HBV DNA (≥10^4^ copies/mL for HBeAg-negative patients, ≥10^5^ copies/mL for HBeAg-positive). According to disease indications and patients’ willingness, patients were suggested to receive oral NA treatment, IFNα treatment, or combination therapy of NAs and IFNα. For oral NA treatment, lamivudine (LAM, 100 mg), telbivudine (LdT, 600 mg), entecavir (ETV, 0.5 mg), or adefovir dipivoxil (ADV, 10 mg) was given daily. Regardless of the presence of cirrhosis, patients were suggested a lifelong NA treatment. Adding-on or switch-to ADV was adopted as rescue therapy for patients with HBV virologic breakthrough under LAM, LdT, or ETV treatment, and vice versa. HBV virologic breakthrough was defined as an increase in serum HBV DNA of 2 log_10_ copies/mL above nadir or detectable after HBV DNA conversion. IFNα treatment was given to patients with ALT ≤ 10× ULN and total bilirubin < 2× ULN. For IFNα treatment, intramuscular injection of recombinant IFN-α1b (5 MU every other day) or subcutaneous administration of pegylated IFNα (Peg–IFN-α2a, 180 mg/week; Peg–IFN-α2b, 1.5 mg/kg body weight/week) was adopted. A 48-week IFNα treatment course was essential for most patients. For those who had partial virologic responses (defined as a <2 log_10_ copies/mL drop) at week 24 of treatment, IFNα treatment course was prolonged to 72 weeks. The IFNα treatment was stopped if there were persistently normal ALT level and undetectable HBV DNA for more than 6 months. Patients with HBV reactivation (HBsAg or HBeAg positive, ALT levels increase, or HBV DNA detectable) were treated again. For patients treated with IFNα plus NAs, NAs treatment was continued after the termination of IFNα treatment. Patients were followed up with regularly at our outpatient clinics or their local hospitals as previously described [17]. The collected follow-up data include antiviral regimens, test results of liver function, medical imaging findings, and occurrence of HCC. The last date of follow-up was 31 August 2019. During the follow-up period, 86 patients were lost to follow-up, 23 patients were diagnosed with HCC within 1 year after collecting baseline information, and 289 patients did not respond to antiviral treatment mostly because they failed to complete antiviral treatment for more than 48 weeks. The above patients were excluded. A total of 2114 patients enrolled in the final analysis.

### 2.3. Serum HBV Genotyping and Viral Mutation Analysis

Peripheral blood samples were obtained at baseline before any clinical treatment. HBV DNA load, serological HBV markers, biochemical parameters for liver function, and platelet count were examined in the study hospitals. Details for viral DNA extraction, HBV genotyping, nested PCR, HBV sequencing (preS region, GenBank accession No. KF167178-KF169170), and viral mutation analysis are provided in the Appendix A. The primers used for nested PCR are listed in Appendix A. Twenty-six preS mutations were evaluated (Appendix A), which were reported to be associated with HCC risk in our previous studies [9,12].

### 2.4. Plasmid Construction

The entire open reading frame (ORF) of preS1/preS2/S (nt.2848-nt.835) was applied for plasmid construction. From HBV-infected patients in this cohort study, we amplified the fragments of wild-type preS1/preS2/S ORF (WT) and the mutant preS1/preS2/S ORFs carrying G2950A/G2951A/A2962G/C2964A (M1), C3116T/T31C (M2), and preS2 deletion (nt.15–nt.56, M3). The fragments of WT, M1, M2, and M3 were inserted into *EcoRI* restriction sites of a *SB* transposon vector, pKT2-FAH-Caggs-SB, which contained the cDNA of *Fah* gene (Appendix A) [19]. For the production of lentiviruses expressing preS1/preS2/S variants, the fragments of WT, M1, M2, and M3 were linked with a flag tag at the 3′ end and were inserted into the plteni-CMV-GFP-puro lentiviral vector. The production of lentiviruses expressing preS1/preS2/S variants and the construction of luciferase reporter plasmid containing the promoter of signal transducer and activator of transcription 3 (STAT3) are detailed in the Appendix A.

### 2.5. Cell Experiments

Huh7 and HepG2 cell lines were purchased from the Chinese Academy of Sciences (Shanghai, China). Before beginning experimentation, all cell lines were authenticated using the genotyping analysis of short tandem repeat (STR) by Biowing Biotechnology (Shanghai, China). All cell cultures were tested for mycoplasma contamination every three months. The cells stably expressing HBV WT-preS1/preS2/S and HBV preS1/preS2/S mutants were constructed with lentivirus. Interleukin-5 (IL-5) and IL-6 were purchased from R&D Systems Inc. (Minneapolis, MN, USA). STAT3 inhibitor Stattic and dimethylsulfoxide (DMSO) were purchased from Sigma (St. Louis, MO, USA). Cell proliferation was assessed by the Cell Counting Kit-8 (CCK8) kit (Dojindo, Osaka, Japan). Migration and invasion were measured using Transwell inserts (Corning, New York, NY, USA). Cell cycle assay and apoptosis assay were performed using flow cytometry (Merck Millipore, Rockville, MD, USA). Cell proliferation, migration, invasion, cell cycle assay, apoptosis assay, luciferase assay, real-time quantitative reverse transcription PCR (qRT-PCR), and Western blot are detailed in the Appendix A. The primers used for qRT-PCR and the antibodies used for Western blot are listed in Appendix A, respectively. The levels of HBsAg in cells and culture medium were determined by ELISA (Jianglai, Shanghai, China). The intracellular HBsAg retention was presented as the ratio of intracellular HBsAg level to the HBsAg level in culture medium. The group of WT was applied as reference to calculate the change fold of intracellular HBsAg retention.

### 2.6. Immunofluorescence Staining

Cells were grown on glass coverslips, fixed in 4% paraformaldehyde, and permeabilized with 0.5% Triton X-100. Fixed cells were blocked with 5% BSA, incubated overnight at 4 °C with primary antibody, washed, incubated with secondary antibody, and counterstained with DAPI (Sigma). Primary and secondary antibodies were listed in Appendix A.

### 2.7. Mouse Models

The *Fah*-deficient mouse was applied to construct the model of the preS1/preS2/S-induced HCC [19]. *Fah*^−/−^ mice suffered lethal liver injury caused by the accumulation of intracellular fumarylacetoacetate. The above phenotype could be rescued through adding 2-(2-nitro-4-trifluoromethylbenzoyl)-1,3-cyclohexanedione (NTBC) to the drinking water of *Fah*^−/−^ mice or exogenously introducing *Fah* gene. In this study, *Fah*^−/−^ mice were maintained with drinking water containing 7.5 μg/mL NTBC. The constructs carrying *Fah* cDNA and the fragment of WT HBV preS1/preS2/S or the fragment with one of the three combo mutations (M1, M2, and M3) (15 μg) were delivered into 5–7-week-old mice by hydrodynamic injection via tail vein. Then, NTBC was removed from the drinking water to select hepatocytes successfully introduced *Fah* cDNA and preS1/preS2/S ORF. The mice that died within 30 days after injection were excluded from analysis due to the failure in gene delivery. The remaining mice were observed for six months to record survival, and they were all sacrificed on Day 180 after the injection. The livers and tumors were subjected to hematoxylin–eosin (H&E) staining and immunohistochemistry (IHC). The cytokine levels in mice plasma were measured by multiplex immunoassay with ProcartaPlex kit (ThermoFisher Scientific, Waltham, MA, USA). For the xenograft experiment, Nod-SCID mice (five-week-old, Jihui Laboratory Animal Care Cooperation, Shanghai, China) were subcutaneously injected with 1.5 × 10^6^ Huh7 cells. Tumors were harvested five weeks later. All animal studies were conducted under the animal welfare protocol approved by the ethics committee of Second Military Medical University.

### 2.8. HBV-Capture Sequencing

Six tissues of the M2-injected *SB* mice were subjected to HBV-capture sequencing, including three liver tissues from tumor-free mice and three tumor tissues from tumor-carrying mice. HBV-capture sequencing was performed with the capture probes of the HBV panel (iGeneTech, Beijing, China) and the HiSeqTM 2500 platform (Illumina, San Diego, CA, USA). Details for DNA extraction, DNA library construction, sequencing, and annotation of HBV integrated breakpoints are described in the Appendix A. HBV-capture sequencing data were uploaded to the Sequence Read Archive (SRA) database under accession number of PRJNA765888.

### 2.9. Gene Expression Profiling Analysis

The gene expression profiles of Huh7 cells were analyzed by RNA-sequencing (RNA-seq). Sequencing libraries were constructed using TruSeq Stranded Total RNA with Ribo-Zero Gold Kits (Illumina, San Diego, CA, USA). The HiSeqTM 2500 sequencing platform (Illumina, San Diego, CA, USA) was applied for RNA-seq. Twelve *SB* mice tissues were subjected to cDNA microarray analysis by using Agilent-074809 SurePrint G3 Mouse GE V2.0 microarray (Agilent Technologies, Santa Clara, CA, USA), including three livers from WT-injected mice and three tumors from each group of preS1/preS2/S mutant-injected mice. Differential gene expression analysis, gene ontology (GO) analysis, and Kyoto Encyclopedia of Genes and Genomes (KEGG) pathway analysis are detailed in the Appendix A. The gene profiling data of *SB* mice and Huh7 cells were uploaded to the Gene Expression Omnibus database (accession number, GSE179125) and the SRA database (accession number, PRJNA762495).

### 2.10. Statistical Analysis

The continuous variables, non-normal data, and categorical variables were compared using Student’s *t*-test, Wilcoxon sum rank test, and Chi-square test, respectively. The Hazard ratio (HR) and 95% confidence interval (CI) were estimated using Cox regression analysis. The significant factors in the univariate Cox analysis were enrolled into the multivariate Cox model. For animal experiments, survival curves were compared by log-rank test. The analysis of the cohort data and experimental data was performed using SPSS (version 18.0, SPSS Inc., Chicago, IL, USA) and GraphPad Prism (version 5.0, GraphPad Software, San Diego, CA, USA), respectively. All statistical tests were two-sided. *p* < 0.05 was considered statistically significant.

## 3. Result

### 3.1. Effects of HBV preS Mutations on the Risk of HCC Occurrence

The 2114 patients were followed up for a median of 11.67 years (IQR, 7.58–15.17 years). Baseline characteristics and follow-up data are shown in Appendix A. In total, 612 patients completed antiviral treatment for ≥48 weeks. Of those, 153 received IFNα therapy alone and 459 received long-term NA treatment. Among those 459 patients, 380 received NAs alone and 79 treated with long-term NAs after the termination of IFNα treatment. The mean duration of long-term NA treatment and IFNα treatment was 111.59 and 68.98 months, respectively. During long-term NA treatment, 150 (32.70%) suffered HBV virologic breakthrough. No significant difference was observed in the rates of G2950A/G2951A/A2962G/C2964A, C3116T/T31C, and preS2 deletion between patients with HBV virologic breakthrough and those without virologic breakthrough (Appendix A). HBV genotype was successfully identified in 1659 patients. HBV genotype C and genotype B accounted for 57.30% and 21.10% of cases, respectively. During 23,845 person-year follow-up, 224 HBV-infected patients developed HCC, with an incidence of 9.39/1000 person-years.

The univariate Cox regression analyses demonstrated that C3116T and T31C were significant risk factors of HCC in all HBV-infected patients, with HRs (95% CI) of 1.70 (1.20–2.40) and 1.60 (1.12–2.28), respectively. Male gender, age, cirrhosis, HBV genotype C, high direct bilirubin, low albumin, high alpha-fetoprotein, and low platelet count were also identified as significant risk factors for HCC. Antiviral treatment was a protective factor, with an HR (95% CI) of 0.54 (0.38–0.76). The preS deletion (pooling all deletion patterns together) was not statistically associated with HCC (data not shown). C3116T and T31C were not identified as independent risk factors by the multivariate Cox analysis (Table 1). Four HBV genotype C-specific mutations, G2950A, G2951A, A2962G, and C2964A, were identified as significant risk factors by the univariate Cox analyses in patients infected with genotype C HBV. Of these four single preS mutations, only A2962G was identified as an independent risk factor in the multivariate Cox model (Appendix A).

The HCC-risk related viral mutations usually existed in the form of combo mutation. Of genotype C HBV-infected patients, 2.91% only carried one single mutation of G2950A, G2951A, A2962G, or C2964A, while 2.62% presented at least two of these four preS mutations, which were defined as combo mutation G2950A/G2951A/A2962G/C2964A positive (Figure 1A). The ratio of patients with one single mutation and those with combo mutation was close. Combo mutation G2950A/G2951A/A2962G/C2964A significantly increased the HCC risk in genotype C HBV-infected patients without antiviral treatment (age- and gender-adjusted HR, 2.49; 95% CI, 1.08–5.72), compared to those with antiviral treatment (Figure 1B). Of all HBV-infected patients, 5.43% presented only one single mutation of C3116T or T31C, whereas 43.61% presented combo mutation C3116T/T31C (Figure 1C). C3116T/T31C significantly increased the HCC risk in patients without antiviral treatment (age- and gender-adjusted HR, 1.61; 95% CI, 1.06–2.45) and was also associated with a trend toward increased HCC risk in patients who received antiviral treatment (Figure 1D). In the preS gene, the region between nt.15 and nt.56 showed the highest deletion frequency (≥7.16%) and was defined as preS2 deletion (Figure 1E). Interestingly, preS2 deletion had no significant effect on the HCC risk of patients without antiviral treatment, but it significantly increased HCC risk in patients receiving antiviral treatment (age- and gender-adjusted HR, 4.11; 95% CI, 1.57–10.72, Figure 1F). We further conducted a stratification analysis based on the types of antiviral treatment. In patients with long-term NA treatment, preS2 deletion has no significant effect on the occurrence of HCC. In patients with IFNα treatment, preS2 deletion was significantly associated with an increased risk of HCC (Appendix A).

### 3.2. Effects of Wild-Type and Mutant preS1/preS2/S on Hepatocarcinogenesis and Inflammation in SB Mice

The empty *SB* vector and constructed plasmids were successfully delivered into 57 *SB* mice. During the six-month observation period, 19 mice died and 15 (78%) developed tumor nodules. M3 was significantly associated with increased mortality of the *SB* mice (Figure 2A). No control mice developed tumor nodules in liver. Compared to WT-injected mice, all the preS1/preS2/S mutant-injected mice displayed a trend toward a higher tumor burden (Figure 2B). The *SB* system-induced integration sites were examined by HBV-capture sequencing in M2-injected mice, because the incidence of tumor was the highest (70%) in M2-injected mice among the groups. It was found that the preS1/preS2/S fragments were mainly integrated in intergenic region (39.36%) and intron region (35.94%). The number of integration sites was similar between tumor tissues and nontumoral liver tissues (Figure 2C). Only three integrated genes were repeatedly detected in different samples, including microtubule associated serine/threonine kinase family member 4 (*Mast4*), transmembrane protein 178B (*Tmem178b*), and zinc finger BTB domain containing 20 (*Zbtb20*). These three genes all had a low ratio of integrated reads to normal reads (0.006–2.41%, Appendix A). Thus, tumor nodules in the *SB* mice should not be caused by insertional mutations. The structure of cancer nests and extensive inflammatory cell infiltration were observed in the tumor tissues from the preS1/preS2/S mutant-injected mice (Figure 2D). The inflammatory pathological changes including inflammatory cell infiltration were more severe in the tissues of the preS1/preS2/S mutant-injected mice than in those of WT counterpart-injected mice. The expression of HBV S protein, cytokeratin 18 (CK18, a classic HCC biomarker), and Ki67 (a biomarker of cell proliferation) was confirmed by IHC in tumors (Figure 2D). The serum levels of three T helper 1 (Th1) cytokines (IFNγ, IL-1β, and IL-12), one Th2 cytokine (IL-5), and IL-6 were significantly upregulated in the preS1/preS2/S mutant-injected mice, compared to the WT-injected counterparts; however, this difference was not observed in the serum levels of tumor necrosis factor α (TNFα), transforming growth factor β (TGFβ1), and vascular endothelial growth factor (VEGF) (Figure 3A). The serum level of IL-12 was significantly higher in the tumor-bearing mice than in the tumor-free mice; moreover, the tumor-bearing mice also displayed a trend toward higher serum levels of IFNγ, IL-5, and IL-6 (Figure 3B). No significant difference was observed in the serum levels of TNFα, IL-1β, TGFβ1, and VEGF between tumor-free mice and tumor-bearing mice (Appendix A).

### 3.3. Effects of HBV preS1/preS2/S Mutants on Malignant Phenotypes of Cancer Cells

Compared to the WT counterpart, stable ectopic expression of M2 and M3 significantly increased the proliferation and migration of Huh7 cells (Figure 4A–C). In HepG2 cells, stable ectopic expression of three preS1/preS2/S mutants all significantly increased cell proliferation and only M3 showed a positive effect on cell migration (Figure 4A–C). None of the preS1/preS2/S mutants showed an obvious effect on cell cycle of Huh7 and HepG2 cells (Appendix A). Ectopic expression of M2 and M3 significantly decreased the percentage of apoptotic cells in Huh7 and HepG2 cell lines, compared to the WT counterpart (Appendix A). The ex vivo experiments showed that ectopic expression of M3 significantly increased the weight of xenograft tumors of Huh7 cells, compared to the WT counterparts (Figure 4D). Ectopic expression of M1 and M2 in Huh7 also displayed a trend toward increasing the weight of xenograft tumors.

### 3.4. Potential Function of HBV preS Mutations

Based on RNA-seq data of Huh7 cells and cDNA microarray data of tissues from *SB* mice, we identified the differentially expressed genes (DEGs, *p* < 0.05, fold change ≥ 2) in groups of the preS1/preS2/S mutants vs. the WT counterpart, respectively. More preS1/preS2/S mutant-induced DEGs (*n* = 1377) were identified in the *SB* mice than in Huh7 cells (*n* = 440) (Figure 5A). Among the top 30 DEGs specifically identified in mice, 12 reported to be involved in different cancers were selected for qRT-PCR validation in Huh7 cells. However, none of these 12 genes were found to be significantly affected by ectopic expression of preS1/preS2/S mutants in vitro. Interestingly, of the 12 genes, five were significantly upregulated by stimulation of inflammatory factors. ETS homologous factor (*EHF*) and phosphoglycerate dehydrogenase (*PHGDH*) were significantly upregulated by IL-5 while laminin subunit gamma 2 (*LAMC2*) and mucin 13 (*MUC13*) were significantly upregulated by IL-6 (Appendix A). Wnt family member 4 (*WNT4*) was upregulated by both IL-5 and IL-6 (Figure 5B). These data suggest that the preS mutation may partially regulate the expression of HCC-related genes by enhancing inflammation rather than directly affecting transcription. We conducted a bioinformatics analysis to further validate this hypothesis. The preS1/preS2/S mutant-induced DEGs identified specifically in the *SB* mice (defined as mice-DEGs) and those identified commonly in both *SB* mice and Huh7 cells (defined as common-DEGs) were subjected to functional analysis, respectively. By GO analysis, ten functional terms associated with inflammation and immune responses were identified in mice-DEGs, whereas only two were identified in common-DEGs (Figure 5C). Among the other GO terms identified in this study, all of four STAT-related terms, three of four ER stress-related terms, and both hypoxia terms were only identified in common-DEGs rather than in mice-DEGs (Figure 5D). KEGG analysis found that most (9/11) of the metabolism-related pathways significantly enriched in the common-DEGs, rather than in the mice-DEGs (Figure 5E).

### 3.5. STAT3 Participated in the preS1/preS2/S Mutant-Induced Hepatocarcinogenesis

STAT3 is involved in the development of many cancers. Compared to WT, ectopic expression of M2 and M3 significantly upregulated the protein level of STAT3 in both Huh7 and HepG2 cells, whereas ectopic expression of M1 only upregulated the protein level of STAT3 in Huh7 cells (Figure 6A). The protein levels of STAT3 and IL-6, the activator of STAT3 pathway, were significantly higher in the tissues adjacent to tumor from M2- and M3-injected mice, compared to their expression in the livers of WT counterpart-injected mice (Figure 6B,C). However, ectopic expression of the preS1/preS2/S mutants had no effects on the mRNA level and the promoter activity of *STAT3* (Appendix A). As mentioned above, stable overexpression of M2 and M3 significantly promoted the proliferation of both HepG2 and Huh7 cells. These effects were significantly attenuated by the STAT3 pathway inhibitor, Stattic (Figure 6D).

### 3.6. HBV preS1/preS2/S Mutants Enhanced ER Stress

We then examined the expression levels of four downstream molecules of ER stress, namely endoplasmic reticulum oxidoreductase 1 alpha (ERO1L), protein kinase R-like ER kinase (PERK), C/EBP homologous protein (CHOP), and X-box binding protein-1 (XBP1) [20,21]. The PERK/CHOP axis and XBP1 both control the cellular responses to ER stress via regulating transcription [22]. ERO1L promotes the disulfide bond formation of proteins. Compared to the WT counterpart, M2 and M3 significantly upregulated all four ER stress molecules in Huh7 cells, whereas M1 only upregulated ERO1L and XBP1 (Figure 7A). ERO1L and PERK were further selected for the validation in tissues from the *SB* mice. The protein levels of ERO1L and PERK were significantly higher in tumors from M2 and preS2 deletion groups, compared to their expression in liver tissues of WT-preS1/preS2/S-injected mice (Figure 7B,C). The ER stress-related pathological change, ground-glass hepatocytes (GGH) [22], was more obvious in liver tissues from each group of the preS1/preS2/S mutant-injected *SB* mice, compared to the WT counterpart-injected *SB* mice (Figure 7C). The retention of HBsAg in ER was a major reason for ER stress. The effects of preS mutations on HBsAg retention were firstly evaluated by ELISA. In comparison with the WT counterpart-expressing Huh7 cells, the ratio of HBsAg level in cells to that in medium was significantly higher in the M2- or M3-expressing Huh7 cells (Figure 7D). Immunofluorescence staining was further performed. In Huh7 cells expressing the WT counterpart, diffuse distribution of HBsAg and calnexin, an ER marker, was observed. However, the staining of HBsAg and calnexin showed a coarse and granular clustering pattern in the perinuclear area of cells expressing the preS1/preS2/S mutants. The co-localization of coarse dot-like cytoplasmic staining of HBsAg and calnexin was observed (Figure 7E). Thus, HBV preS mutants, especially M2 and M3, may enhance the retention of HBsAg in ER.

## 4. Discussion

In this study, we demonstrated that antiviral treatment was not only an independent protective factor of HCC in HBV-infected patients but also influenced the effects of HBV preS mutations on the HCC risk. Combo preS mutations G2950A/G2951A/A2962G/C2964A and C3116T/T31C significantly increased the HCC risk only in HBV-infected patients without antiviral treatment. Interestingly, HBV preS2 deletion significantly increased the risk of HCC in HBV-infected patients with IFNα treatment. The C-to-T mutation in the HBV enhancer I/X promoter region was reported to affect the response of HBV to interferon therapy [23,24]. For the first time, this study identified the statistical association between preS2 deletion and the effect of IFNα on cancer prevention. The relevant mechanism should be further investigated. Compared to the WT counterpart, HBV preS1/preS2/S with preS2 deletion induced a higher tumor burden in the *SB* mice, increased cell proliferation in vitro, and promoted xenograft tumor growth. These results are consistent with previous studies demonstrating the oncogenic effect of HBV preS deletion in cell models, HBV transgenic mouse models, and the models of HBV infection [9,22,25]. For the first time, we found that G2950A/G2951A/A2962G/C2964A and C3116T/T31C combo mutations displayed a similar effect of preS2 deletion on carcinogenesis. Compared to the WT counterpart, M1 and M2 induced a trend toward a higher tumor burden in the *SB* mouse models and xenograft models of Huh7 cells. M1 promoted the proliferation of HepG2 cells and M2 promoted the proliferation of both Huh7 and HepG2 cells. These preS combo mutations, especially C3116T/T31C, may act as pro-oncogenic molecules through a similar mechanism by which HBV preS1/preS2/S with preS2 deletion induces carcinogenesis.

We found that HBV preS2 deletion and preS combo mutations promote hepatocarcinogenesis via deteriorating the inflammatory microenvironment. The serum levels of IL-5 and IL-6 were significantly upregulated in the preS1/preS2/S mutant-injected mice, compared to the WT counterpart-injected mice. In addition, the preS1/preS2/S mutants induced dramatic alteration in the transcriptome of tissues from the *SB* mice, an animal model with an intact immune system. Much fewer DEGs were observed in the WT-expressing cells vs. the preS1/preS2/S mutant-expressing cells. Furthermore, some DEGs specifically identified in the *SB* mice, such as *WNT4*, were only upregulated in Huh7 cells by the stimulation of IL-5 and/or IL-6, rather than by ectopic expression of the preS1/preS2/S mutants. Therefore, HBV preS mutations might regulate the transcription of oncogenes indirectly via altering the inflammatory microenvironment in humans with an intact immune system. The high-risk preS mutations accumulated during hepatocarcinogenesis can lead to immune escape by inducing the loss of B-cell epitopes or T-cell epitopes in HBV envelope proteins [26]. In HBV-infected patients with normal immunity, the deletion rate of the preS2 at nt.15–nt.56 region, which codes the B-cell epitopes, is higher than that of other preS2 regions. In immune-suppressed kidney-transplant patients with HBV infection, the deletion rate of nt.15–nt.56 is lower than that of other regions [27]. Interestingly, our data suggest that the inflammation-selected preS mutations could deteriorate the inflammatory microenvironment in turn. In a study of HBV preS1/preS2/S with preS2 deletion transgenic mice, the mRNA levels of pro-inflammatory (TNFα, IL-6, IL-1α, and IL-1β) and chemoattractant cytokines were significantly upregulated in the liver. The increased proliferation of leukocytes is also reported [25]. HCC patients with HBV preS2 deletion in the circulation display a higher number of regulatory T cells (Tregs) and a higher expression level of immune checkpoint molecule programmed death ligand 1 (PD-L1) in tumor tissues than do patients without preS2 deletion [28,29]. Both increased Tregs and PD-L1 are associated with a poor prognosis of HCC [30,31]. Therefore, the feedback between inflammation and the preS mutations contributes simultaneously to the immune tolerance of mutant HBV antigens and the maintenance of pro-oncogenic inflammation.

This study also found that HBV preS mutations promote hepatocarcinogenesis via activating the STAT3 pathway and ER stress. The activation of the IL-6/STAT3 pathway is the hallmark events of hepatocarcinogenesis [20]. The effects of HBV preS mutations on the protein level of STAT3 may be due to the aberrant ER function. ER stress is activated in response to an accumulation of unfolded or misfolded proteins, such as mutated HBV surface proteins, in the ER lumen. In this study, M2 and preS2 deletion significantly enhanced the retention of HBsAg in ER, compared to the WT counterpart. Sustained activation of ER stress endows malignant cells with greater tumorigenic, metastatic, and drug-resistant capacity [32]. The ER stress sensors, PERK and XBP1, were both reported to promote the survival capacity of cells by activating STAT3 [33,34]. PERK was also activated during the process of epithelial-to-mesenchymal transition [35]. M2 and M3 significantly upregulated the protein level of PERK and XBP1, whereas M1 significantly upregulated XBP1. M1, M2, and M3 all significantly upregulated the protein level of another ER stress molecule, ERO1L, in cells and the *SB* mice. The PERK/CHOP axis, ERO1L, and XBP1 promote the tolerance of cells to the hypoxia microenvironment and alter the metabolism function of mitochondria [20,21,36,37,38]. Consistent with the results of these previous studies, the GO term KEGG pathways associated with hypoxia and metabolism were significantly enriched in the preS mutation-induced DEGs identified commonly in the *SB* mice and Huh7 cells. The preS mutation-activated ER stress may also promote pro-oncogenic inflammation. ERO1L and XBP1 can activate IL-6 pathway and promote the progression of lung cancer and HCC, respectively [39,40]. Thus, the preS mutations promote hepatocarcinogenesis possibly via inducing pro-oncogenic inflammatory cytokine and signaling pathways associated with metabolism, the two major pathways related to ER stress.

The preS region of HBV harbors a complex spectrum of viral mutations. Compared to preS2 deletion, the oncogenic effect of base substitution in the preS gene was underestimated and the underlying mechanism was not investigated. We, for the first time, provide evidence supporting the oncogenic function of C3116T/T31C and G2950A/G2951A/A2962G/C2964A combo mutations. Remarkably, C3116T/T31C had a stronger capacity for upregulating all the four investigated molecules of ER stress in Huh7 cells, compared to HBV preS2 deletion. The prevalence of C3116T/T31C (43.61%) was higher than that of preS2 deletion (7.16%). Thus, this combo mutation has potential clinical implications for a wider population of HBV-infected patients.

This study evaluated the prevalence and oncogenic effects of preS mutations in Chinese who were infected with HBV genotypes B and C. There are studies reported the prevalence of preS mutations in European who were infected with HBV genotypes A, D, and E. An Italian study identified preS2 deletion in 7.5% HBV infected patients and preS/S single base mutations in 25% HBV infected patients [41]. A Swedish study identified preS2 mutations in 50% HBV-HCC patients and in 21% non-HCC HBV infected patients [42]. Interestingly, a Spanish study reported that the rate of preS deletion was significantly higher in sub-Saharan patients than in Caucasian patients [43]. The effects of preS mutations on HCC in patients from different races should be further investigated.

Our study has limitations. First, we did not experimentally verify the effect of HBV preS2 deletion on drug resistance. Second, cDNA microarray analyses were conducted in the liver tissues from WT-preS1/preS2/S-injected mice and the liver tissues with tumors from the preS1/preS2/S mutant-injected mice, because it is difficult to separate fresh liver tissues from the tumor tissues of the preS1/preS2/S mutant-injected mice.

## 5. Conclusions

HBV combo mutations C3116T/T31C and G2950A/G2951A/A2962G/C2964A significantly increased the risk of HCC in patients without antiviral treatment, whereas the preS2 deletion (nt.15–nt.56) significantly increased the risk of HCC in patients who received antiviral treatment. The preS1/preS2/S mutants can promote hepatocarcinogenesis via inducing ER stress, altering metabolism, and pro-oncogenic STAT3 inflammatory pathways. HBV-infected patients with HCC-risk preS mutations will benefit from treatments targeting the IL-6/STAT3 pathway and ER stress-related pathways.

## Figures and Tables

**Figure 1 cancers-14-03274-f001:**
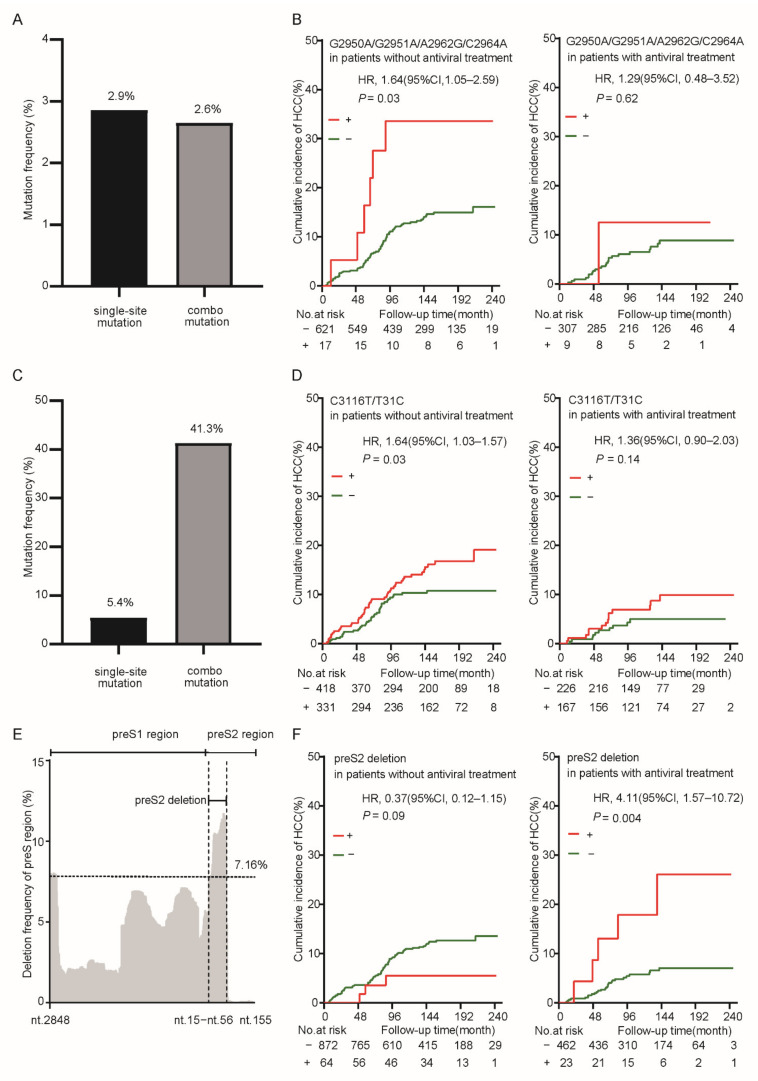
The impact of baseline preS mutations on HCC occurrence in a prospective cohort study with 2114 HBV-infected patients. (**A**) The frequencies of G2950A, G2951A, A2962G, and C2964A single mutations and G2950A/G2951A/A2962G/C2964A combo mutation. Patients with at least two of these four mutations were defined as combo mutation positive. (**B**) The effect of G2950A/G2951A/A2962G/C2964A combo mutation on HCC occurrence in genotype C HBV-infected patients without antiviral treatment (left) and in those who received antiviral treatment (right). (**C**) The frequencies of C3116T and T31C single mutations and C3116T/T31C combo mutation. (**D**) The effect of HCC C3116T/T31C combo mutation on HCC occurrence in all HBV-infected patients without antiviral treatment (left) and in those who received antiviral treatment (right). (**E**) The deletion frequency of each locus in the preS1/preS2 region. The region with the highest deletion rate (nt.15–nt.56) was defined as the preS2 deletion in this study. (**F**) The effect of preS2 deletion on HCC occurrence in HBV-infected patients without antiviral treatment (left) and in those who received antiviral treatment (right). *p* values were calculated by cox regression analysis.

**Figure 2 cancers-14-03274-f002:**
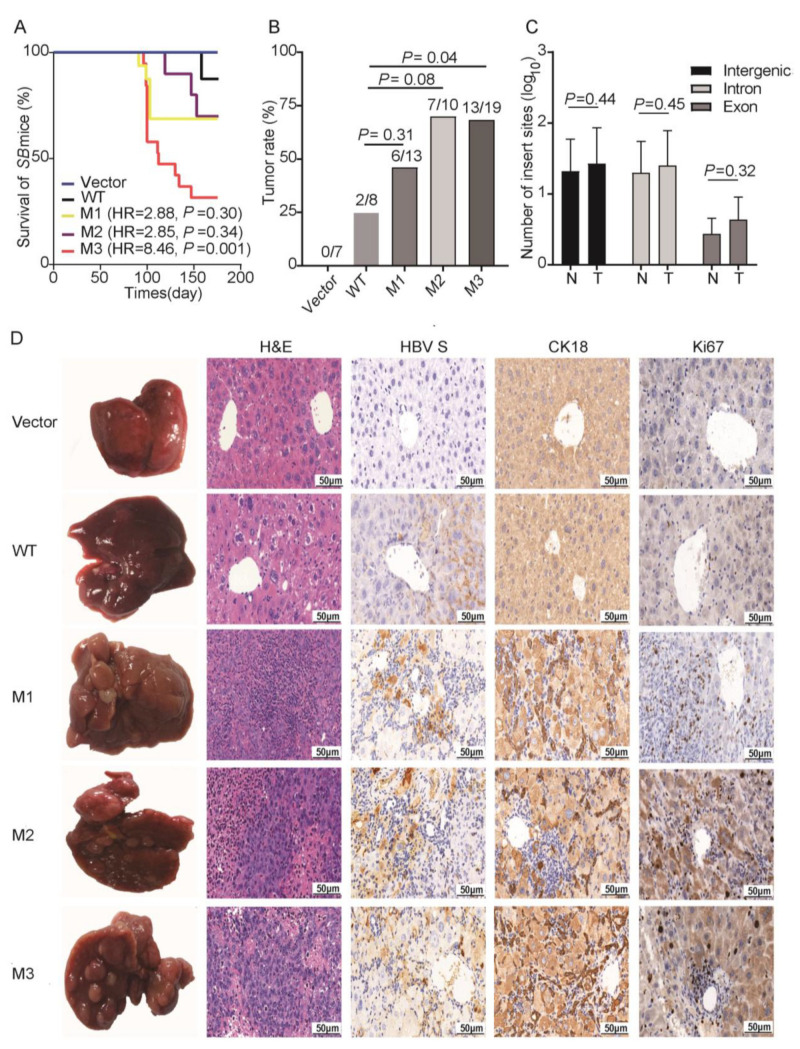
The survival, tumor occurrence, HBV DNA insertion sites, and pathological characteristics of the *Sleeping Beauty* (*SB*) mouse models. (**A**) The survival of *SB* mice during six-month observation. Vector, WT, M1, M2, and M3 indicate the mouse models injected with the empty vector, wild-type preS1/preS2/S, G2950A/G2951A/A2962G/C2964A, C3116T/T31C, and preS2 deletion, respectively. *p* values were calculated by cox regression analysis. (**B**) The rate of tumor occurrence in *SB* mouse models, including all the survival mice and the dead mice during the observation period. *p* values were calculated by Student’s *t*-test. (**C**) The number of HBV preS1/preS2/S insertion sites in different functional regions and different tissue types. T, tumors from the mice with tumor nodules; N, liver tissues from the tumor-free mice. *p* values were calculated by Wilcoxon sum rank test. (**D**) Representative images of H&E and IHC staining of the livers (from Vector and WT groups) and tumors (from M1, M2, and M3 groups). Column 1, representative gross features; Column 2, representative H&E staining; Columns 3–5, representative IHC staining for HBV S protein, CK-18, and Ki67, respectively. *p* values were calculated by Student’s *t*-test.

**Figure 3 cancers-14-03274-f003:**
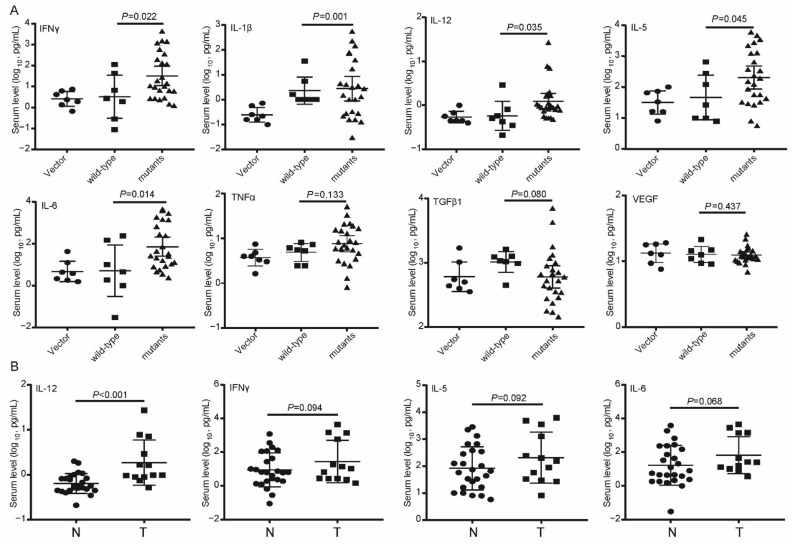
The expression levels of cytokines in mice serum. (**A**) The serum levels of IFNγ, IL-1β, IL-12, IL-5, IL-6, TNFα, TGFβ1, and VEGF in the SB mice injected with empty vector, wild-type preS1/preS2/S fragment, or preS1/preS2/S mutants. (**B**) The serum levels of IL-12, IFNγ, IL-5, and IL-6 in the SB mice with tumor (T) or those without tumor (N).

**Figure 4 cancers-14-03274-f004:**
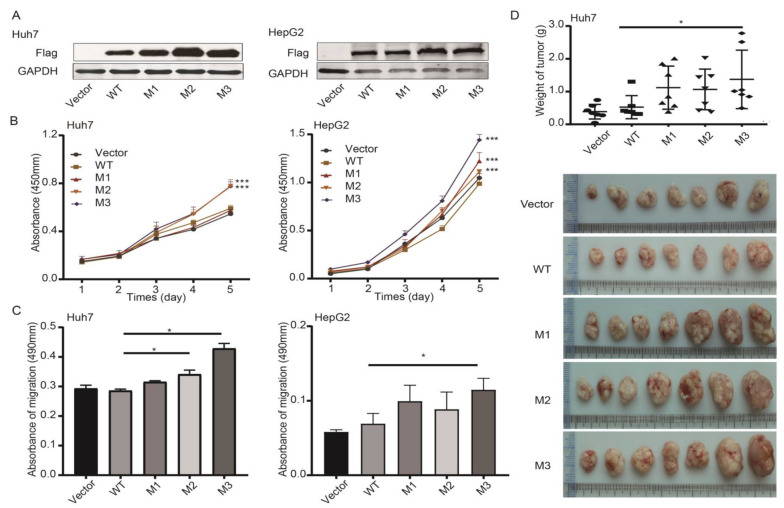
Effects of the preS1/preS2/S mutations on malignant phenotypes of cancer cells. (**A**) Ectopic overexpression of HBV preS1/preS2/S mutants and WT counterpart was confirmed by detecting Flag tag in Huh7 (**left**) and HepG2 cells (**right**). (**B**) Effects of preS mutations on the proliferation of Huh7 (**left**) and HepG2 cells (**right**). (**C**) Effects of preS mutations on the migration of Huh7 (**left**) and HepG2 cells (**right**). (**D**) The weight of xenograft tumors among different groups. * *p* < 0.05, *** *p* < 0.001.

**Figure 5 cancers-14-03274-f005:**
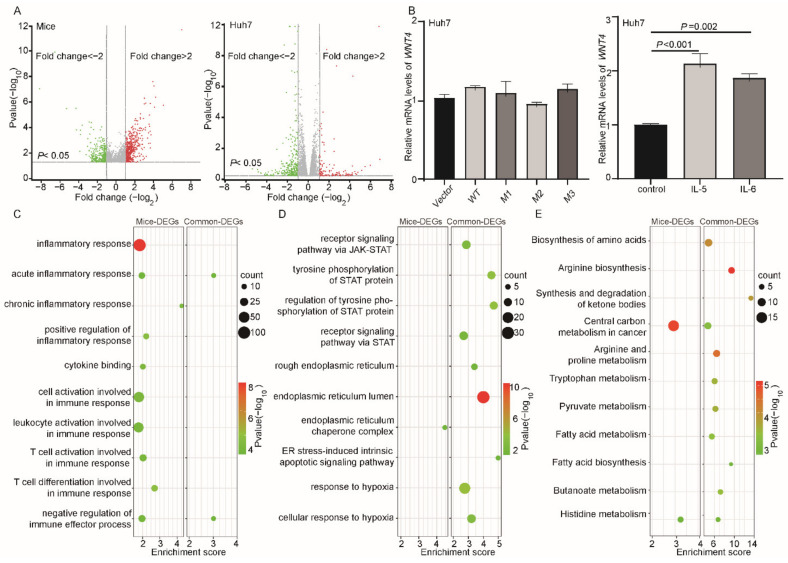
Functional analysis of HBV preS mutations. (**A**) (**left**) Differential expressed genes (DEGs) in preS1/preS2/S mutant-injected *SB* mice vs. WT preS1/preS2/S-injected *SB* mice (*n* = 1377). (**right**) DEGs in HBV preS1/preS2/S mutant-expressing Huh7 cells vs. HBV preS1/preS2/S WT-expressing Huh7 cells (*n* = 440). *p* values and fold changes were calculated by nbinom Test. (**B**) The mRNA levels of *WNT4* in Huh7 cells which is detected by qRT-PCR. (**left**) Ectopic expression of HBV preS1/preS2/S mutants has no direct effect on the mRNA expression of *WNT4*. (**right**) Effects of IL-5 and IL-6 (100 ng/mL, 1 h), two preS1/preS2/S mutant-induced cytokines, on the mRNA expression of *WNT4*. *p* values were calculated by Student’s *t*-test. (**C**) Most of the inflammation- and immune response-related GO terms were significantly enriched in DEGs that were identified specifically in *SB* mice (mice-DEGs) rather than in DEGs identified both in *SB* mice and Huh7 cells (common-DEGs). (**D**) ER stress-, STAT pathway-, and hypoxia-related GO terms were significantly enriched in common-DEGs, rather than in the mice-DEGs. (**E**) Most of the KEGG identified metabolism-related pathways significantly enriched in common-DEGs, rather than in the mice-DEGs.

**Figure 6 cancers-14-03274-f006:**
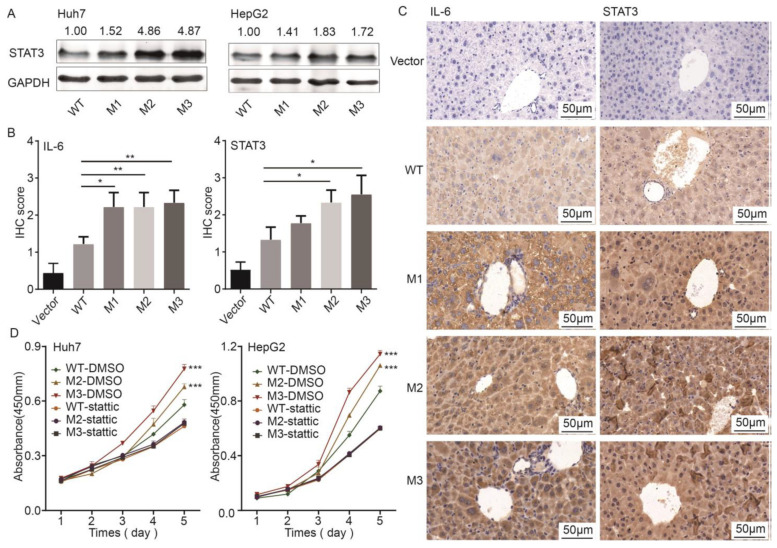
The roles of STAT3 in the preS1/preS2/S mutant-induced hepatocarcinogenesis. (**A**) The protein level of STAT3 was higher in Huh7 and HepG2 cells with ectopic expression of preS1/preS2/S mutants than in those with ectopic expression of the WT counterpart. (**B**) The histograms displaying the IHC scores of IL-6 and STAT3 in the liver tissues (vector group and WT group) and tissues adjacent to tumors (groups of preS1/preS2/S mutants) from *SB* mice. (**C**) Representative images for the IHC staining of IL-6 and STAT3 in adjacent liver tissues to tumors. (**D**) The effects of mutant HBV S proteins on the cell proliferation in Huh7 and HepG2 cells. In cells treated with 0.1% DMSO, M2 and M3 significantly increased the proliferation of HepG2 and Huh7 cells, compared with WT. In cells treated with 10 μM Stattic, no significant difference was observed in the proliferation among cell expressing M2, M3, and WT. * *p* < 0.05, ** *p* < 0.005, *** *p* < 0.001.

**Figure 7 cancers-14-03274-f007:**
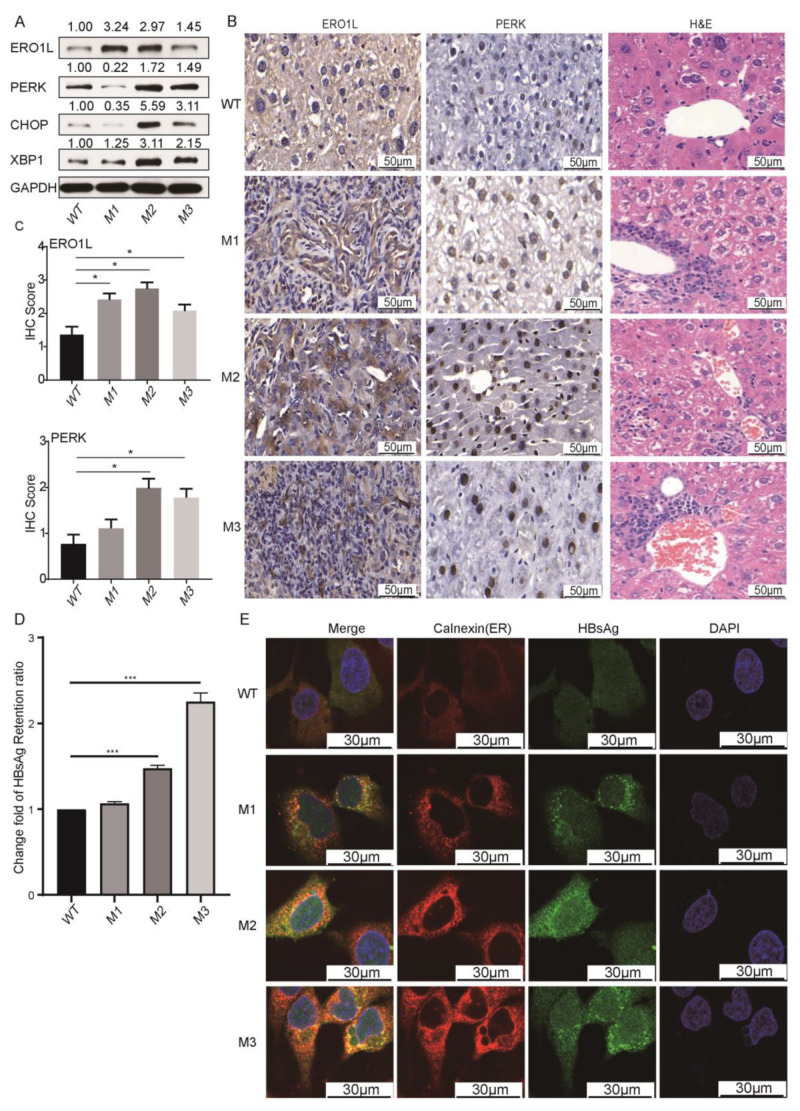
The preS1/preS2/S mutants significantly enhanced ER stress and the retention of HBsAg in ER. (**A**) The protein levels of ERO1L, PERK, CHOP, and XBP1 in Huh7 cells with ectopic expression of WT or preS1/preS2/S mutants. (**B**) Representative IHC and H&E images. Columns 1 and 2 show the IHC staining of EROL1 and PERK. In column 3, the ER stress-related pathological change, ground-glass hepatocytes, was more obvious in liver tissues from each group of preS1/preS2/S mutant-injected *SB* mice. (**C**) The histograms displaying the IHC scores of EROL1 and PERK in the liver tissues (vector group and WT group) and tumor tissues (groups of preS1/preS2/S mutants) from *SB* mice. (**D**) The change fold of intracellular HBsAg retention. The level of HBsAg in cells and medium was evaluated by ELISA. The HBsAg was presented by the ratio of intracellular HBsAg level to the HBsAg level in medium. The WT group was applied as reference to calculate change fold. (**E**) The result of immunofluorescence staining. Huh7 cells with the stable expression of WT, M1, M2, or M3 were immunostained with mouse anti-HBsAg (green) and rabbit anti-calnexin antibodies (red), with DAPI (blue) as nuclear counterstain. Co-localization of red and green immunofluorescence is visualized as yellow in merge images (first column) * *p* < 0.05, *** *p* < 0.001.

**Table 1 cancers-14-03274-t001:** Univariate and Multivariate Cox Regression Analysis of Factors Significantly Affected the Occurrence of Hepatocellular Carcinoma.

Variable	No. (%) of Participants (*n* = 2114)	Person-Years of Follow-Up	No. of HCC (*n* = 224)	Incidence Rate per 1000 Person-Years	Univariate Analysis HR (95% CI)	*p* Value	Multivariate Analysis HR (95% CI)	*p* Value
Gender								
Female	490 (23.18)	5987	30	5.01	1		1	
Male	1624 (76.82)	17,858	194	10.86	2.08 (1.41–3.05)	<0.001	3.28 (1.78–6.04)	<0.001
Age (years)								
<60	1885 (89.17)	21,363	190	8.89	1		1	
≥60	229 (10.83)	2482	34	13.70	1.62 (1.12–2.33)	0.010	1.92 (1.15–3.20)	0.013
Antiviral treatment								
No	1502 (71.05)	17,361	184	10.60	1		1	
Yes	612 (28.95)	6484	40	6.17	0.54 (0.38–0.76)	<0.001	0.61 (0.39–0.96)	0.031
Cirrhosis								
No	1624 (76.82)	19,372	126	6.50	1		1	
Yes	490 (23.18)	4473	98	21.91	3.39 (2.60–4.41)	<0.001	2.04 (1.25–3.32)	0.004
C3116T								
C	755 (35.71)	8403	56	6.66	1		1	
T	587 (27.77)	6765	75	11.09	1.70 (1.20–2.40)	0.003	1.24 (0.55–2.81)	0.600
T31C								
T	675 (31.93)	7605	51	6.71	1		1	
C	608 (28.76)	6898	73	10.58	1.60 (1.12–2.28)	0.011	0.97 (0.43–2.22)	0.95
HBV genotype								
B	447 (21.14)	4977	27	5.42	1		1	
C	1212 (57.33)	13,887	137	9.87	1.86 (1.23–2.81)	0.003	1.38 (0.79–2.42)	0.26
Direct bilirubin (mol/L)								
≤7	691 (32.69)	8370	56	6.69	1		1	
>7	1396 (66.04)	15,157	167	11.02	1.63 (1.21–2.21)	<0.001	0.92 (0.58–1.46)	0.71
Albumin (g/L)								
≥35	1335 (63.15)	16,255	113	6.95	1		1	
<35	772 (36.52)	7507	109	14.52	2.08 (1.59–2.70)	<0.001	1.29 (0.80–2.07)	0.30
A-fetoprotein (ng/mL)								
≤20	1358 (64.24)	16,014	124	7.74	1		1	
20–400	752 (35.57)	7791	100	12.84	1.61 (1.23–2.09)	<0.001	1.39 (0.93–2.07)	0.11
Platelet count (10^9^/L)								
100–300	1004 (47.49)	12,050	68	5.64	1		1	
<100	1110 (52.51)	11,795	156	13.23	2.35 (1.77–3.12)	<0.001	1.22 (0.77–1.91)	0.39

## Data Availability

The cDNA microarray data of *SB* mouse models were uploaded to the Gene Expression Omnibus (GEO) database under accession number of GSE179125. The HBV-capture sequencing data of the *SB* mice and the RNA-sequencing data of Huh7 cells were uploaded to the Sequence Read Archive (SRA) database under accession numbers of PRJNA765888 and PRJNA762495, respectively.

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
