# Peer review of "HBV preS Mutations Promote Hepatocarcinogenesis by Inducing Endoplasmic Reticulum Stress and Upregulating Inflammatory Signaling"

_cancers, 2022, doi:10.3390/cancers14133274_

Round 1

Reviewer 1 Report

Liu et colleagues evaluated 2114 HBV-infected Chinese patients , of whom 612 who received antiviral treatments, enrolled in 1998-2007, and followed-up to 2019, to see the association with preS mutations and HCC onset.

However:

  • presence of alcoholic hepatitis / concomitant use was not excluded
  • HBV DNA was expressed in copies/ml (according to year) but only 1/3 of patients treated with antivirals: levels of HBsAg or HBeAg status or ALT levels should be added in methods section.
  • the type of antiviral therapy has not been described, and this is important for conclusions of the study. Lamivudine? entecavir? how many patients developed resistance to lamivudine? how many HBV flares secondary to it? how many HCC in this population? 
  • no description of mean duration of antiviral treatment (but only ">48 months"), but it's different according to IFN (finite time) or NA use (theorically indefinite or long-term deruation), and both drugs had different antiviral effects, and IFN cannot be used in decompensated patients. Any HBV flares after the finite treatment of NA?
  • only patients under effective long-term NA treatment could be the control group to evaluated the effects of pres2 mutations and HCC onset
  • conclusions ("hbv patient under antiviral should be monitored for generation of pre-s2 deletion") cannot be accepted in the present form
  • any references evaluating this mutation in caucasian population, to add in discussion?

Author Response

Response:

Authors are grateful to the comments of this reviewer. The comments are very important to improve the manuscript.  We revised manuscript accordingly. Details were listed in the file "Response to reviewer 1". Please see the attachment.

Best regards,

Guangwen Cao, the corresponding author

Reviewer 2 Report

Liu et al. Investigate the correlation between HBV preS mutations or deletions and hepatocarcinogenesis. To this aim, they analyze a cohort of 2114 CHB patients for the distribution of specific preS mutations and preS deletions (analyzed as a pool) and their contribution as risk factors for developing HCC. The results were different according to the mutation type, HBV genotype and presence or not of antiviral treatment, but overall strongly indicated a causal relationship between preS mutations and HCC occurrence.

To go further in the mechanisms underlying the link between preS mutations/deletions and HCC pathogenesis, the authors used in vitro models and an in vivo fah-deficient mouse model. After expression of preS mutants/deletions, RNA-Seq, immunohistochemistry and cytokine assessment, among other experiments, demonstrated that preS mutants/deletions caused significant alterations of gene expression profiles, ER-stress, cell metabolism and upregulation of STAT3 signaling pathway. Interestingly, the differences between the gene expression pathways alterated in vivo and in vitro suggest a prominent role of the host immune response in preS mutants/deletions-associated hepatocarcinogenesis.

The manuscript is clear, despite being very long, particularly in the Discussion section. The data have the added value of including a CHB cohort and both in vitro and in vivo models. The major limitations of the work consist in:

1) the overall messages are not new: there is a huge literature on preS mutations/deletions and HCC pathogenesis, involving ER stress, cytokine profile deregulation and STAT3 pathway;

2) the analysis of the patients’ cohort raises major concerns: firstly, how is it possible that NUC-treated patients still have >6 log of viremia (Table S4). Secondly, no data is provided on the type of NUC used for therapy, this is crucial to understand a possible link between preS mutations and resistance to treatment. In this respect, the speculation of the authors in the Discussion section go too far, since they do not provide any experimental evidence.

3) the mouse model is not clear to me: the construct used contains the fah cDNA and the “fragment WT, M1, M2 or M3”, what does it mean specifically? Is the entire S ORF there also or only the preS region?

4) The most interesting data of the paper are the differences between in vitro and in vivo gene regulation and the possibility that this would be due to a role of host immune response (not present in vitro). Unfortunately, the authors do not develop this aspect. For example, some immunostaining for immune infiltrates in mice liver would have been informative.

5) the data in figure 4B,C and 6B,D are not clear: how do they measure cell growth? How could it be an increase in cell growth without cell cycle perturbation? 

Author Response

(The authors gave the same response as above.)

Round 2

Reviewer 1 Report

No further comments after the improvement and corrections

Reviewer 2 Report

the authors fully replied to the raised concerns